# Study and Optimization Defect Layer in Powder Mixed Electrical Discharge Machining of Titanium Alloy

Dragan Rodic *, Marin Gostimirovic, Milenko Sekulic, Borislav Savkovic and Andjelko Aleksic

Department of Production Engineering, Faculty of Technical Sciences, University of Novi Sad,
21000 Novi Sad, Serbia; maring@uns.ac.rs (M.G.); milenkos@uns.ac.rs (M.S.); savkovic@uns.ac.rs (B.S.);
andjelkoa94@uns.ac.rs (A.A.)
* Correspondence: rodicdr@uns.ac.rs

**Abstract:** Electrical discharge machining (EDM) has recently become very popular for processing titanium alloys, but surface quality is a major problem. During machining, a defect layer inevitably forms on the surface, which can have a negative impact on surface quality. One of the ways to reduce the defect layer is to add powder to the dielectric. However, it is not yet completely clear which powder and in what quantity it should be added to reduce the defect layer. In this sense, the present study aims to investigate the effects of machining parameters on the defect layer in powder-mixed electrical discharge machining of titanium alloys. The main goal is to achieve the minimum thickness of the defect layer by optimally adjusting the input parameters. Experimental studies were performed using the Taguchi orthogonal array $L_9$, considering discharge current, pulse duration, duty cycle, and graphite powder concentration as input parameters. Based on the Taguchi and ANOVA analyses, the discharge current was found to have the greatest effect on the defect layer. In addition, analysis of variance revealed that pulse duration was the second influential parameter, followed by graphite powder and duty cycle. The minimum thickness of the defect layer is obtained at a discharge current of 1.5 A, a pulse duration of 30 μs, a duty cycle of 50%, and a graphite powder concentration of 12 g/L. The results obtained in this study provided answers to some of the unresolved research questions and confirmed the findings that the proposed method can be applied in the industry.

**Keywords:** defect layer; discharge current; pulse duration; duty cycle; graphite powder; Taguchi



## 1. Introduction

Electrical discharge machining (EDM) is based on the removal of material by a series of repetitive electrical discharges between electrodes (tool and workpiece) in the presence of a dielectric fluid. All electrically conductive materials can be machined by this process. However, its use is most justified in the machining of high-alloy steels, hard metals, and metal–ceramics [1]. Besides the basic advantages, such as machining complex surfaces, inaccessible surfaces, etc., EDM also has its disadvantages. Extremely high temperatures occur in the work area during EDM, so the occurrence of thermal defects in the surface layer of the workpiece material (microstructural changes, residual stresses, microcracks, etc.) is to be expected [2].

During EDM, dielectric fluid is constantly introduced into the machining zone. This causes a sudden cooling of the upper surface of the workpiece. At the same time, the material not removed from the machining zone solidifies at high speed due to the high thermal conductivity of the dielectric. In this way, a recast layer is formed. It is usually fine-grained, brittle, and hard, i.e., it has a different microstructure than the original material. Below the recast layer, a heat-affected zone is created due to the high temperature discharged (plasma zone) [3]. The molten layer and the heat-affected zone also form the defect layer (*DL*) during EDM. In general, the formation of the layer depends primarily on the processing conditions and then on the properties of the workpiece (chemical composition and thermal

conductivity). These are undesirable phenomena that must be minimized or removed after machining.

To minimize the defect layer, the EDM process was renewed and modified. The addition of electrically conductive powder to the dielectric creates a modified material removal process known as Powder Mixed Electrical Discharge Machining (PMEDM) [4]. The addition of graphite powder changes the mechanism of EDM machining. Powder added to the liquid dielectric reduces the insulation properties and causes an increase in the working gap between the tool and the workpiece [5]. An increase in the working gap means more efficient circulation of the dielectric, i.e., a washout of the working space between the tool and the workpiece [6]. In this way, EDM becomes more stable, which improves the technological characteristics of the process [7]. In addition, the heat in the machining zone is reduced, which has a positive effect on the defect layer [8].

In recent years, researchers around the world have conducted many studies and obtained good results in effectively reducing the defect layer generated during EDM. Muthuramalingam and Phan [9] studied the defect layer during the PMEDM process. Here, the effects of input parameters on the formation of defect layers during silicon steel processing were analyzed. It was found that PMEDM produces a smaller thickness of the defect layer with a uniform distribution over the machined surface. Ahmed et al. [10] also studied the effects of powder mixing parameters on the defect layer during electrical discharge machining with copper and graphite electrodes. The lowest defect layer values of 5.0 μm and 5.57 μm are obtained at high current and low current with low pulse duration using the copper and graphite electrodes and silicon carbide powder, respectively. The analysis of the effect of adding graphene oxide in the dielectric was carried out by Świercz et al. [11]. The main objective was to minimize the defect layer of chromium steel during EDM. The best properties of the defect layer are obtained with 0.1 percent graphene oxide in the dielectric with negative polarity. Xu et al. [12] presented studies in which mixed boron carbide powder was used in the processing of titanium alloys. A significant reduction in the defect layer and better surface quality was achieved. To investigate the machining performance of titanium alloy PMEDM, Shahriar et al. [13] studied the influence of two types of powders: graphite and titanium oxide. The influence of different discharge currents and graphite concentrations on the heat-affected zone was analyzed. The best results were obtained at a concentration of 7 g/L. The presented studies show the importance of research related to the addition of powder to the dielectric. The question at which powder concentration the defect layer can be minimized is still controversial.

To find the optimal powder concentration, some researchers have used modern optimization methods. Various methods were used, such as Taguchi's approach, the response surface method, etc. [11,14,15]. By applying the aforementioned methods, optimal solutions were usually found, where the minimum thickness of the defect layer is achieved [16,17]. Tripathy et al. [18] studied the effects of input parameters such as graphite powder concentration, discharge current, pulse duration, duty cycle, and gap voltage on metal removal rate, tool wear, surface roughness, and defect layer thickness in the PMEDM of die steel. The Taguchi method was used to determine the optimal parameters at which the thickness of the defect layer is the smallest. To solve the multicriteria decision-making problem, Phan [19] used the Taguchi method. One of the initial features was a defect layer. Optimal parameters were obtained when the defect layer was 8.56 μm. Hosni et al. [20] optimized the input parameters of discharge current, pulse duration, and chromium powder concentration to achieve the minimum thickness of the defect layer. By using the response surface method, the minimum thickness was achieved at a discharge current of 20.12 A, a pulse duration of 50.14 μs, and a powder concentration of 3.96 g/L.

A review from previous researchers [10–19] found that very few studies have reported the effects of adding graphite powder on the thickness of the defect layer during the PMEDM of titanium alloys. This can be explained by the fact that titanium alloys are unfavorable for EDM machining in terms of surface quality [21,22]. This is because some titanium alloys contain admixtures of aluminum, which have a negative effect on the

machining process [23]. Therefore, this research addresses the problem of optimizing input parameters such as discharge current, pulse duration, duty cycle, and graphite powder concentration. The discharge current and pulse duration have the greatest influence on the formation of the defect layer and directly determine the discharge energy, which also depends on the amount of heat in the processing zone. Pulse duration and pulse off time are used to calculate the duty cycle, which also affects the amount of energy during the pause time. In addition to the parameters that can be set on the machine, one of the non-electrical parameters that must be considered in PMEDM analysis is the powder concentration. The discharge energy is also affected by the concentration of graphite powder. If it is in the machining zone, the gap distance increases, as this leads to an earlier electrical discharge, which means a lower amount of heat on the surface of the workpiece. Obtaining information about the defect layer condition is a lengthy and expensive process. Therefore, the Taguchi orthogonal array $L_9$ is used in this study. Four input parameters on three levels were used. The thickness of the defect layer, i.e., the thickness of the recast layer and the heat-affected zone, was chosen as the output parameter, one of the indicators of surface integrity.

The main objective and contribution of this study is the selection of optimal parameters that allow for achieving the minimum thickness of the defect layer in PMEDM titanium alloys. The importance of the presented research work can be seen in the fact that it contributes to the development and application of EDM from two aspects. The first aspect concerns research work in academic circles. The results of this research contribute to a detailed analysis of the latest methods of EDM of titanium alloys. The second aspect is the possible application in industry. The application of the presented principles gives a more detailed picture of the innovative application of EDM from the aspect of the defect layer. For the purpose of quality control, a single-criteria optimization was performed, and the verification of the assumed optimal parameters of the PMEDM titanium alloy was carried out, resulting in a minimum thickness of the defect layer. The importance of this study is reflected in the answering of some unresolved research questions regarding the amount of graphite powder added and confirms the results that the proposed method can be applied in industry.

## 2. Materials and Methods

### 2.1. General Machining Conditions

A series of experiments were performed on an Agie Charmilles type SP1-U die-sinking EDM machine (Beijing Agie Charmilles Industrial Electronics Co., Ltd., Beijing, China). The isotropic graphite with a cross section of $10 \times 10$ mm$^2$ was used as an electrode for machining titanium alloy TiAl$_4$V$_6$. The dielectric Ilocut EDM 180 (Castrol Industrial, Cambridge, UK) was used for the experimental studies. Additives such as graphite powder and surfactants (reagents) are added to the dielectric to better flush the machining zone and prevent particle agglomeration (shrinkage and accumulation). During the machining, the additives added to the machining zone reduce the insulating properties of the dielectric, increasing the working distance. Asbury PM19 graphite powder (Asbury Graphite & Carbons NL B.V., Maastricht, The Netherlands) was used in the PMEDM of titanium alloys. Tween 20 C$_{58}$H$_{114}$O$_{26}$ surfactant (Biosolve Chimie, Dieuze, France) is a transparent liquid with high density. The role of the surfactant is to prevent shrinkage or accumulation of graphite powder particles to ensure a homogeneous mixture of powder and dielectric during PMEDM.

### 2.2. Powder Mixed Electrical Discharge Machining

There are two types of PMEDM systems: open and closed. In the open PMEDM process, there is a standard dielectric circuit system with special filters [5]. The basic component of the closed PMEDM system is the working tank, which is placed as a modular part in the working area of the classical EDM machine. For the needs of dielectric eroding with mixed powders, a tank with elements for fixing and positioning the closed workpiece

was designed and manufactured, Figure 1. In this way, contamination of the machine tool with graphite powder is prevented, which leads to minimization of costs. The dimensions of the tank are 330 × 330 × 330 mm, with a capacity of 20 L. With such an adapted system, it is necessary to ensure the proper distribution of the powder, as well as the cleaning of the work area.

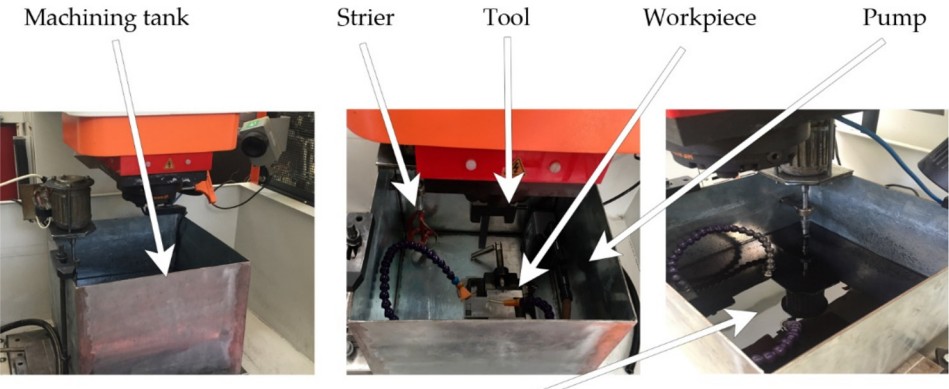

**Figure 1.** Setup of PMEDM.

### 2.3. Systematized Selected PMEDM Parameters

The parameters affecting the output of EDM of titanium alloys can be divided into two groups: parameters of electric pulse and non-electric parameters of the process. The systematized conditions for the processing of titanium alloys are listed in Tables 1 and 2.

**Table 1.** Machining conditions.

| Parameters of EDM | Label | Value | Units |
|---|---|---|---|
| Discharge current | $I_e$ | 1.5 to 7.5 | A |
| Pulse on time | $t_i$ | 24 to 240 | μs |
| Pulse off time | $t_o$ | 24 to 240 | μs |
| Open circuit voltage | $U_0$ | 100 | V |
| Polarity | *Pol* | (-) | / |
| Duty factor | $\tau$ | 30 to 70 | % |

**Table 2.** Non-electrical parameters.

| Non-Electrical Value | Symbol | Value | Unit |
|---|---|---|---|
| Retract distance | *UP* | 1.5 | mm |
| Erosion time | *DN* | 2 | s |
| Graphite powder | *GR* | 0 to 12 | g/L |
| Surfactant | *SR* | 10 | g/L |
| Dielectric flow | *Q* | 20 | L/min |
| Machining time | *T* | 60 | min |

The discharge current is limited by the dimensions of the front surface of the electrode (tool), i.e., the current density. According to the recommendations of the literature [24,25], the maximum current density for graphite tools during roughing is in the range of 10 ÷ 20 A/cm$^2$, depending on the type of paired materials. In order to determine the upper limit of the discharge current, an experiment was conducted with a current of 9.5 A. From Figure 2, it can be seen that the surface is damaged and of very poor quality. Therefore, this study used a discharge current in the range of 1.5 to 7.5 A, taking into account the possible variations at the machine tool. According to research [24,25], the upper limit of pulse duration for machining titanium alloys is 200 to 500 μs. Since the research is

based on the minimization of the defect layer, the upper limit was set at 240 μm. According to the literature [24], a value of the duty factor higher than 50% has a negative effect on the defect layer of the titanium alloy [26–28]. For this reason, the duty factor was varied within the limits of 30 to 70%. The concentration of graphite powder usually ranges from 0 to 20 g/L for different paired materials [29–31]. To determine the upper limit of graphite powder concentration, the test was performed with a powder concentration of 20 g/L. The surface of the workpiece was damaged. Then, the powder concentration was varied in a range of 0 to 15 g/L.

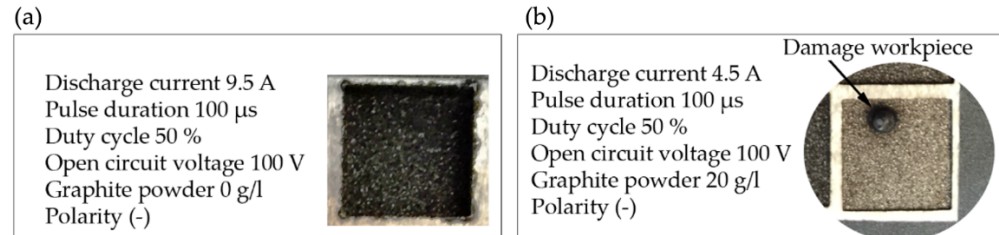

(a)

Discharge current 9.5 A
Pulse duration 100 μs
Duty cycle 50 %
Open circuit voltage 100 V
Graphite powder 0 g/l
Polarity (-)

(b)

Damage workpiece

Discharge current 4.5 A
Pulse duration 100 μs
Duty cycle 50 %
Open circuit voltage 100 V
Graphite powder 20 g/l
Polarity (-)

**Figure 2.** Preliminary experiments: (**a**) determination of the upper limit of the discharge current and (**b**) determination of the upper limit of the graphite powder concentration.

In performing the experiments, a side wash with dielectric was used with a flow rate of 20 L/min through a nozzle with a diameter of 4 mm and another nozzle with a cross section of $2 \times 8$ mm. The tool lift-off time was 2 s, and the distance was 1.5 mm. The erosion time for each test point was 60 min.

*2.4. Experiment Plan*

The Taguchi plan is a method that allows for reducing the number of experimental points by orthogonal arrangements and minimizing the effects outside the influential parameters [32]. It was applied in this study with the aim of more accurately optimizing and analyzing the influence of input parameters on surface integrity. That is, a defect layer consisting of a recast layer and a heat-affected zone. In the experimental study of the PMEDM process for titanium alloys, the following input parameters were chosen: discharge current ($I_e$), pulse duration ($t_i$), duty cycle ($\tau$), and graphite powder concentration ($GR$). The processing parameters and values are listed in Table 1. An experimental design according to the Taguchi orthogonal array $L_9(3^4)$ with four factors and three levels was established. The orthogonal array consists of four columns (factors) and one row (experimental points), Table 3.

**Table 3.** Taguchi orthogonal array $L_9(3^4)$ at PMEDM TiAl$_6$V$_4$.

| No. | Factor | | | | Defect Layer |
|---|---|---|---|---|---|
| | $I_e$ (A) | $t_i$ (μs) | $\tau$ (%) | $GR$ g/L | $DL$ (μm) |
| 1. | 1.5 | 32 | 30 | 0 | 6.51 |
| 2. | 1.5 | 75 | 50 | 6 | 6.54 |
| 3. | 1.5 | 180 | 70 | 12 | 6.92 |
| 4. | 3.2 | 32 | 50 | 12 | 9.33 |
| 5. | 3.2 | 75 | 70 | 0 | 10.52 |
| 6. | 3.2 | 180 | 30 | 6 | 11.31 |
| 7. | 6.0 | 32 | 70 | 6 | 12.60 |
| 8. | 6.0 | 75 | 30 | 12 | 11.51 |
| 9. | 6.0 | 180 | 50 | 0 | 13.20 |

*2.5. Defining the Thickness of the Defect Layer*

Since PMEDM involves extremely high temperatures in the machining zone, the occurrence of thermal defects in the surface layer of the workpiece material is to be expected.

Since the above changes affect the condition of the surface layer, which can significantly affect the function of the part, it is necessary to be particularly attentive. To measure the thickness of the defect layer, an optical microscope (Leitz Orthoplan light microscope, Oberkochen, Germany) with a maximum magnification of up to $500\times$ was used.

The thickness of the defect layer was determined by taking three measurements at the points where the thickness was greatest. The defect layer consists of a recast layer and a heat-affected zone [33,34]. An example of the determination of the thickness of the defect layer in PMEDM of titanium alloys is shown in Figures 3 and 4.

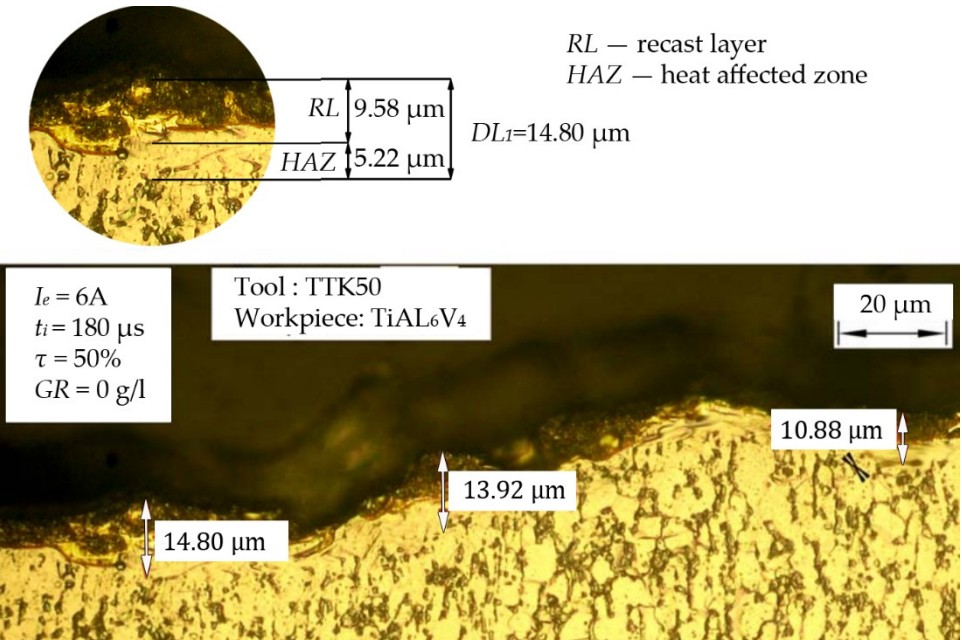

**Figure 3.** Thickness of the defect layer for the first measurement.

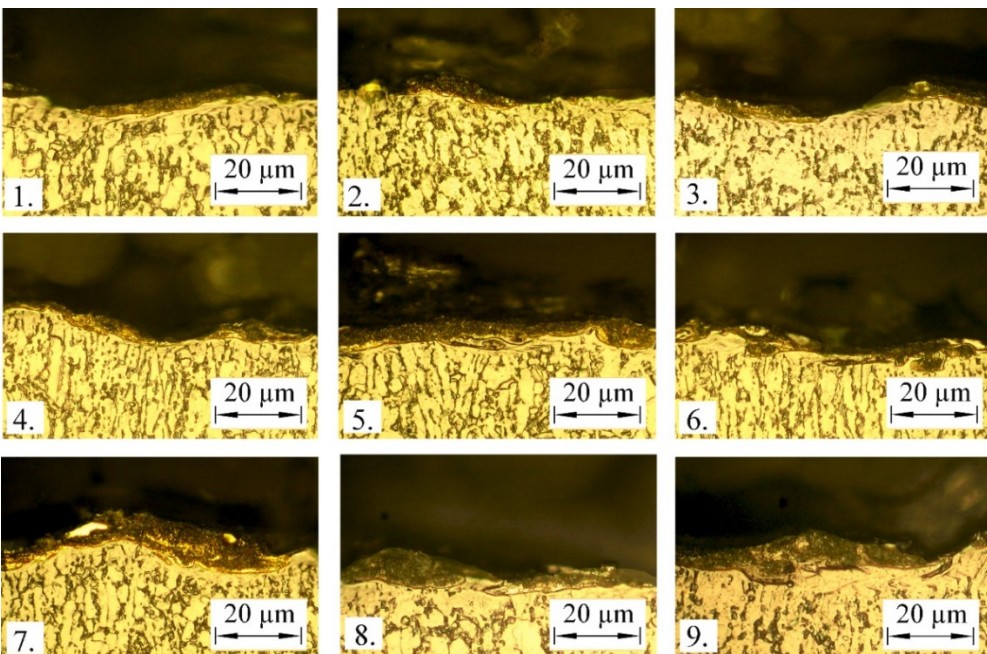

**Figure 4.** Defect layers for all experiments by numbers according to Table 3.

## 3. Results and Discussion

For single-objective optimization of the PMEDM parameters, the Taguchi method was used, in which the thickness of the defect layer was optimized in experiments based

on the orthogonal Taguchi array $L_9(3^4)$. This method does not require the creation of a mathematical model and represents an alternative approach for identifying the optimal input parameters. The advantages of the Taguchi method are reflected in a simple and systemically efficient approach to optimization [35,36]. This approach is reflected in the optimization problem, where the logarithmic function of the observed response is used as the objective optimization function, known as the signal-to-noise *S/N*.

Optimization involves static problems in which several controlled input parameters determine the value of the output variable. To determine the influence of each input variable on the observed response, it is necessary to calculate the signal-to-noise (*S/N*) ratio. The signal-to-noise ratio is the measure for the analysis and evaluation of experimental results. Based on the *S/N* ratio, the effect of a change in the input parameters on the output value can be estimated.

Based on the measured values of the defect layer, the *S/N* ratio for all nine experiments was calculated based on the Taguchi "small is better" quality feature. This feature is applied in the case where the minimum response value is required, Equation (1) [37]:

$$(S/N)_j = -10 \log \frac{1}{n} \sum_{i=1}^{n} \left[ y_{ij}^2 \right], \; j = 1 \dots m \tag{1}$$

where $n$ is the number of experiments and $y_{ij}$ is the measured value of the output.

Depending on the consideration of the type of output parameters, different approaches of the Taguchi method are used. For each selected type of output value, a higher *S/N* ratio represents a better result. The analysis of the results of the experiment according to the Taguchi plan is mainly performed by a series of statistical calculations, such as calculations of the mean effects of the factor levels and the optimal conditions, and the estimation of the output variable at the optimal level.

The average effect of factor A at level 1 for the orthogonal Taguchi series $L_9(3^4)$ is calculated according to Equation (2) [37]:

$$\bar{A}_1 = \frac{y_1 + y_2 + y_3}{3} = \frac{y_{A_1}}{3} \tag{2}$$

Equation (2) shows an example of calculating the mean effect of factor $A$ at the first level, varying four factors at three levels. Here, $y_1$, $y_2$, and $y_3$ are the results of the experiment for which factor $A$ has the first level. Similarly, the average factor effect can be calculated for other input parameters. The average of all experiment results or the mean of the measured output powers for the orthogonal array $L_9(3^4)$ is calculated according to Equation (3) [37]:

$$\bar{y} = \frac{y_1 + y_2 + \dots + y_9}{9} \tag{3}$$

The contribution of each input factor, when set to the desired level, represents the difference between the average effect of the factor and the average total output. An example of calculating the contribution of a factor when set at level 1 can be represented by Equation (4) [37]:

$$A = \bar{A}_1 - \bar{y} \tag{4}$$

The estimate of the output size at the optimal values of the input parameters is obtained by adding all the contributions of the individual input factors to the mean value of the output powers. If the optimal combination of input parameters for the observed output characteristic of the $L_9(3^4)$ orthogonal array is $A = 1$, $B = 2$, and $C = 3$, the expected optimal values can be calculated according to Equation (5) [37]:

$$y_{opt} = \bar{y} + \left( \bar{A}_1 - \bar{y} \right) + \left( \bar{B}_2 - \bar{y} \right) + \left( \bar{C}_3 - \bar{y} \right) \tag{5}$$

By using the optimal levels of the input factors, the expected *S/N* ratio of the optimal levels can be calculated according to Equations (6) and (7) [37]:

$$S/N_{opt} = \overline{S}/N + \sum_{i=1}^{p} \left( S/N_{iopt} - \overline{S}/N \right) \qquad (6)$$

$$\overline{S}/N = \frac{1}{n}\sum_{i=1}^{n} S/N_i \qquad (7)$$

where the variables are as follows:

*S/N$_{opt}$*—*S/N* ratio of the i-th factor at the optimal level.

$\overline{S}/N$—the total value of the *S/N* ratio.

*p*—the number of factors influencing the performance characteristic.

*S/N$_i$*—*S/N* ratio in the ith experiment.

The expected output value at the optimal level can be calculated based on the expected *S/N* ratio at the optimal level, depending on which approach is chosen. An example of calculating the output at the optimal level for the "small is better" approach is shown in Equation (8) [37].

$$y_{opt} = 10^{\frac{-S/N_{opt}}{20}} \qquad (8)$$

Table 4 shows the *S/N* ratios with each factor and the corresponding level for the thickness of the defect layer. The factors with the largest difference in mean values (max–min) have the greatest influence on the output size. From the table, it can be seen that the discharge current has the greatest influence on the defect layer, followed by the pulse duration, the concentration of graphite powder, and finally, the duty cycle.

**Table 4.** Response table of *S/N* ratio for defect layer.

| | Factors | Levels | | | Min-Max | Rang |
|---|---|---|---|---|---|---|
| | | 1 | 2 | 3 | | |
| 1. | **(A) Discharge current** | −16.46 | −20.3 | −21.88 | 5.42 | 1 |
| 2. | **(B) Pulse duration** | −19.23 | −19.32 | −20.09 | 0.87 | 2 |
| 3. | **(C) Duty cycle** | −19.52 | −19.37 | −19.75 | 0.38 | 4 |
| 4. | **(D) Graphite powder** | −19.71 | −19.8 | −19.14 | 0.66 | 3 |

The influence of individual input parameters on the output power of the PMEDM can be illustrated with the help of a response diagram showing the change in the *S/N* ratio at the moment of changing the level of the control parameter from 1 to 3. Accordingly, the influence of the parameters on the output is graphically expressed by the slope angle of the line connecting the different parameter levels [38]. Observing the slope of the lines, you can see that the steepest line for the factor is A, then B, then D, and finally C. This order of responses corresponds to the calculated rank (Table 4). According to Figure 5, the highest *S/N* ratio indicates the optimal level of each factor. Therefore, based on the "smaller is better" criterion, the optimum combination of the PMEDM titanium alloy input parameters as a function of the minimum thickness of the defect layer is A = 1, B = 1, C = 2, and D = 3.

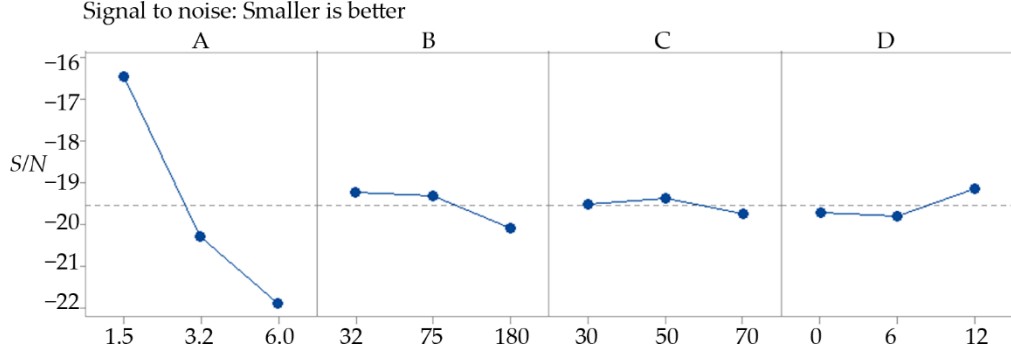

**Figure 5.** Graphic representation of the *S/N* ratio for thickness of the defect layer, (**A**) Discharge current, (**B**) Pulse duration, (**C**) Duty cycle and (**D**) Graphite powder.

The optimization of the PMEDM parameters of the titanium alloy, which allows the optimal value for the thickness of the defect layer, is shown in Table 5. The optimum value for the thickness of the defect layer is *DL* = 5.99 μm for the obtained *S/N* ratio of −15.56.

**Table 5.** Optimal setting of input parameters with confirmation experiment.

| Input | Level | Value | Obtained *DL* Using the Taguchi Method | Confirmation Experiment |
|---|---|---|---|---|
| $I_e$ **(A)** | 1 | 1.5 | *S/N* = −15.56 | *DL* = **6.32 μm** |
| $t_i$ **(μs)** | 1 | 32 | | |
| **τ (%)** | 2 | 50 | *DL* = 5.99 μm | |
| **GR (g/L)** | 3 | 12 | | |

The confirmation test is the final step to verify the improvement in the defect layer at the optimal level of process parameters. The average error between the values of the EDM output power obtained by the prediction based on the Taguchi analysis and the values obtained after the verification experiments (with the optimal input parameter values) was only 5.22%. According to the research of [39–41], a prediction is considered good if the average error is up to 10%. Therefore, the single-objective optimization of the input parameters of the PMEDM can be considered successful. However, the proposed principle is only applicable in the context of the experiment, i.e., within the domains of the input factors. In order to achieve greater applicability of the proposed method, it is necessary to expand the range of input parameters as well as a larger amount of experimental data.

The usual way to analyze and summarize the results is the analysis of variance (ANOVA) table. ANOVA is a statistical technique used to evaluate the relative importance of each process factor [42]. The main goal of the ANOVA is to obtain an answer from the results of the experiment as to how much the variation of each factor affects the overall variation of the observed outcome.

According to Taguchi plan L$_9$(3$^4$), based on the ANOVA performed via the F-test, it is possible to see the influence, i.e., the percentage participation of each factor in the PMEDM titanium alloy, on the thickness of the defect layer. Factors whose F value is less than 1 were excluded from the analysis, which was the case for the coefficient of impulse effect (factor C) [38]. After the exclusion of the insignificant factors, the ANOVA for the remaining members is presented in the reduced Table 6, where the percentage participation for factors A, B, and D is given. The discharge current has the greatest influence on the thickness of the defect layer, with a percentage of 93.53%. The percentage of 3.46% is taken by the pulse duration for the set process conditions, while the concentration of graphite powder influences only 2.68%. The presented ANOVA analysis confirms the results obtained with the Taguchi method.

**Table 6.** Reduced ANOVA table based on the Taguchi plan.

| Source | DF | Sum Sq | Mean Sq | F-Value | Percent % |
|---|---|---|---|---|---|
| **A—$I_e$** | 2 | 51.5238 | 25.7619 | 306.57 | 93.53 |
| **B—$t_i$** | 2 | 1.9041 | 0.952 | 11.33 | 3.46 |
| **D—$GR$** | 2 | 1.4873 | 0.7436 | 8.85 | 2.68 |
| **Error** | 2 | 0.1681 | 0.084 | | 0.33 |
| **Total** | 8 | 55.0832 | | | |

A similar study was conducted by Kolli and Kumar [30], in which the discharge current, surfactant concentration, and graphite powder concentration were varied. Analysis of ANOVA revealed that discharge current (81.83%) was the most important factor compared to other factors such as surfactant concentration (5.97%) and graphite concentration (10.53%). The same conclusions were reached by the research of [43,44], in which a high percentage of the proportion of discharge current was associated with discharge energy. Higher discharge current leads to an increase in discharge density and a decrease in washing efficiency. The higher discharge energy leads to higher melting temperature, increased evaporation, and high impulse forces acting on the machined zone, which affects the higher non-uniform defect layer formed on the top of the workpiece surface. When analyzing pulse duration [13], it generally ranks second in terms of influence. This is explained by the fact that it directly affects the discharge energy [45]. The pulse duration controls the time that the current is allowed to flow per cycle, and it is directly proportional to the amount of thermal energy supplied during the discharge period. In general, the concentration of the powder has less influence than the discharge current and pulse duration [21]. Nevertheless, analyses show that there is a reduction in the defect layer, which is due to the space and time available for interactions between the plasma channel and the powder additives for energy transfer [7]. The graphite powder particles cause the plasma channel to widen during the electrical discharge. This results in vaporization and ionization of the dielectric in this region. The graphite particles at the interface absorb the discharge energy [41]. As a result, a smaller portion of the discharge energy is available to melt the workpiece material, and shallower pools of molten material are formed, resulting in a smaller defect layer.

ANOVA graphs allow visual identification of the influence of input parameters in the interval defined by the experimental space. The graphs of the response ANOVA for the output processing parameters are shown in Figure 6. It is noticeable that the influence of the observed parameters on the thickness of the defect layer is expressed by the angle of inclination of the line connecting each level of the factor. As the intensity of the discharge current increases, the thickness of the defect layer visibly increases.

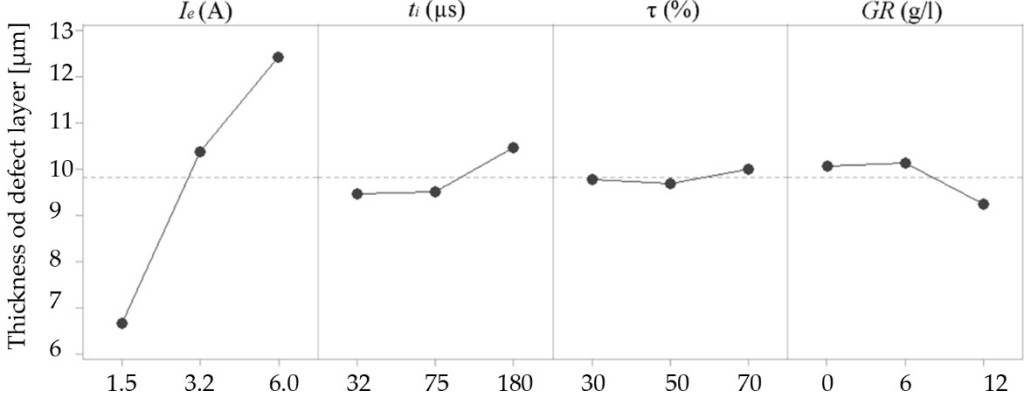

**Figure 6.** Response ANOVA graph for the thickness of the defect layer.

This analysis confirmed the order of influence of input parameters during machining compared to published research on PMEDM titanium alloys [12,46]. In addition to discharge current, which had the greatest influence on the defect layer as expected, pulse

duration, duty cycle, and graphite powder concentration had less influence than expected. From the data presented in this study, it can be concluded that pulse duration has a greater influence on PMEDM titanium alloys at higher discharge currents.

The explanation for the exclusion of the duty cycle from the analysis of ANOVA is due to the fact that this parameter does not have a significant influence on the defect layer at a relatively short pulse duration, up to 180 µs in this study. A significant impact of the duty cycle is expected for values of pulse duration greater than 200 µs, as a higher discharge energy occurs. A higher discharge energy has a detrimental effect on the surface integrity of the machined titanium alloy if the pulse off time is too short (calculated in $\tau > 90\%$), which has been confirmed in studies [47,48].

For the optimal combination of machining conditions, the following machining performances were monitored in addition to the defect layer: surface roughness, material removal, and tool wear rate. Arithmetical mean height ($R_a$) and the maximum height of the profile ($R_{max}$) were monitored as indicators of microgeometry, and values of 1.62 µm ($R_a$) and 10.90 µm ($R_{max}$) were obtained. In the research of [6,30], similar values were obtained for the above parameters. To further reduce surface roughness, the discharge energy must be reduced [8]. This is because a lower discharge energy produces a lower impulse force in the machining zone and influences the formation of small craters, resulting in a high surface finish. Then, a value of 0.42 mm$^3$/min was obtained for the material removal rate (MMR). The explanation for the low value is that the optimum combination is tuned to reduce the defect layer, and a low discharge current of 1.5 A was used. According to investigations [19,24], the discharge current has the greatest influence on the MRR. When the discharge current increases, MRR also increases [44]. Based on the optimum values adjusted to the minimum thickness of the defect layer, a relative tool wear of 51.23% was obtained. Although a low discharge current of 1.5 A was used, a high percentage of tool wear was obtained, which is not the case for EDM machining of steel [49]. When machining titanium, the TWR decreases as the discharge current increases [50]. This phenomenon is explained by the increase in discharge energy, which leads to the formation of a thicker titanium carbide layer on the surface of the workpiece. The formed layer of titanium carbide directly affects the reduction in relative tool wear [24]. It can be concluded that other parameters (MRR and TWR) related to processing costs may deteriorate when PMEDM is based on an output parameter, which in this case is the defect layer. The MRR is directly related to the processing time, and the TWR to the cost of the tool [48]. It is known that in some cases the cost of the tool represents up to 50% of the total machining cost [51]. Therefore, it is necessary to pay special attention to all parameters in the multi-objective optimization.

The innovative development directions of EDM, as well as the optimization methodology presented in this study, have elevated EDM to a higher level, especially from the aspect of surface integrity. In this context, the objectives of this work have been fully achieved.

## 4. Conclusions

In order to minimize the thickness of the defect layer of a titanium alloy produced by EDM and improve the quality of the machined surface, this study proposed adding graphite powder to the dielectric. The result of the present research is extremely helpful for the selection of optimum machining conditions for PMEDM titanium alloy, and the following conclusions can be drawn.

i.      The discharge current is the most important process parameter affecting the defect layer, followed by the pulse duration and the graphite powder concentration;

ii.     The optimal parameter A1B1C2D3 was determined as follows: discharge current 1.5 A; pulse duration 32 µs; duty cycle 50%; graphite concentration 12 g/L; obtaining a minimum defect layer of 5.99 µm; and corresponding *S/N* ratio −15.56;

iii.    The confirmatory experiment resulted in a thickness of 6.32 µm. The average error between the Taguchi analysis and the values obtained after the confirmatory experiments was only 5.22%;

iv. To confirm the previous result, a ANOVA analysis was performed to study the influence of the parameter on the thickness of the defect layer. The results show that the discharge current affects 93.53%, the pulse duration 3.46% and the concentration of graphite powder 2.68%.

The research conducted in this work covers only a small part of the PMEDM field. Future research in the field of PMEDM process improvement and optimization should focus on testing the PMEDM process of titanium alloys considering a larger number and wider intervals of input factors, as well as research for different erosion depths. In addition, the influence of graphite powder granulation may be one of the input factors that could affect the output performance of EDM of advanced engineering materials. In addition to granulation, the shape of graphite powder grains may also be an important factor in better understanding the process. The significance of this research can be seen from two points of view. The first is a better understanding of PMEDM from the point of view of how much graphite powder to add. Second, the verification of the optimal regimes confirms that the Taguchi method can be applied in the industry, but in the context of an experimental design.

**Author Contributions:** Conceptualization, D.R. and M.G.; methodology, D.R., M.S. and B.S.; software, D.R. and A.A.; validation, D.R. and B.S., writing—original draft preparation, D.R.; writing—review and editing, D.R., M.G. and M.S.; visualization, D.R. and M.G., supervision M.S., A.A. and B.S., project administration, D.R., M.G. and A.A. All authors have read and agreed to the published version of the manuscript.

**Funding:** This research received no external funding.

**Data Availability Statement:** Data sharing is not applicable to this article.

**Acknowledgments:** This paper has been supported by the Provincial Secretariat for Higher Education and Scientific Research through project no. 142-451-1772/2022-01/01: "Research on the innovative process of electrical discharge machining of titanium alloy".

**Conflicts of Interest:** The authors declare no conflict of interest.

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
