# Peer review of "Study and Optimization Defect Layer in Powder Mixed Electrical Discharge Machining of Titanium Alloy"

_processes, doi:10.3390/pr11041289_

Round 1

Reviewer 1 Report

In this article, the authors present a powder-mixed electrical discharge machining of titanium alloys. They added graphite powder as dielectric to reduce the defect layer using Taguchi approach. The structure of the work is well prepared. It has a scientific quality. I think the quality of the work can be increased by some of the comments below.

1. In the abstract, the authors should indicate the achieved thickness of the defect layer at optimal parameters.

2. In the introductory part, the last paragraph, you should state the aim of the paper and especially the contribution.

3. Page 5, line 172, check the number of figures. I think we need figures 1 and 2.

4. Page 8, line 279, substitute the abbreviation DDS.

5. In some parts of the text the term "Duty Cycle" is used, in others it is not. It is necessary to agree in the text. For example, line 295.

6. In the discussion section, authors should state that the minimum thickness of the defect layer is reached at a concentration of 12 g/l. And make a comment on it.

Author Response

The authors thank the reviewer very much for the helpful suggestions and comments to improve the manuscript. All changes in the main text are modified. All changes are marked in green in the revised changes document.

Reviewer 2 Report

Please refer to the attached file

Author Response

(The authors gave the same response as above.)

Round 2

Reviewer 2 Report

Congratulations to the authors on their progress with the manuscript. Although many improvements have been made, a few comments still need to be addressed before it can be accepted for publishing. For the next revision, please remove all sentences that authors want to remove, and for any changes, please use the blue colour font.

Please refer to the attached file

Author Response

Dear reviewer, all changes in the text are marked in blue.
We would like to thank you for your specific comments, which have significantly improved our paper.
Yours sincerely!
